# Current Trends and Future Prospects of Molecular Targeted Therapy in Head and Neck Squamous Cell Carcinoma

**DOI:** 10.3390/ijms22010240

**Published:** 2020-12-29

**Authors:** Naoya Kitamura, Shinya Sento, Yasumasa Yoshizawa, Eri Sasabe, Yasusei Kudo, Tetsuya Yamamoto

**Affiliations:** 1Department of Oral and Maxillofacial Surgery, Kochi Medical School, Kochi University, Nankoku, Kochi 783-8505, Japan; shinya-sento@kochi-u.ac.jp (S.S.); yoshizawa@kochi-u.ac.jp (Y.Y.); yoshieri@kochi-u.ac.jp (E.S.); yamamott@kochi-u.ac.jp (T.Y.); 2Department of Oral Bioscience, Tokushima University Graduate School of Biomedical Sciences, 3-18-15 Kuramoto, Tokushima 770-8504, Japan; yasusei@tokushima-u.ac.jp

**Keywords:** head and neck squamous cell carcinoma, molecular targeted therapy, cetuximab, immune checkpoint inhibitor, multi-oncogene panel test, photoimmunotherapy

## Abstract

In recent years, advances in drug therapy for head and neck squamous cell carcinoma (HNSCC) have progressed rapidly. In addition to cytotoxic anti-cancer agents such as platinum-based drug (cisplatin and carboplatin) and taxane-based drugs (docetaxel and paclitaxel), epidermal growth factor receptor-tyrosine kinase inhibitors (cetuximab) and immune checkpoint inhibitors such as anti-programmed cell death-1 (PD-1) antibodies (nivolumab and pembrolizumab) have come to be used. The importance of anti-cancer drug therapy is increasing year by year. Therefore, we summarize clinical trials of molecular targeted therapy and biomarkers in HNSCC from previous studies. Here we show the current trends and future prospects of molecular targeted therapy in HNSCC.

## 1. Introduction

Multidisciplinary treatment for head and neck squamous cell carcinoma (HNSCC) consists of three methods: surgery, anti-cancer drug therapy, and radiotherapy. Among them, the progress of anti-cancer drug therapy is rapid, and the combination of them is diversifying, including the recent molecular targeted therapy by immune checkpoint inhibitor (ICI), and their importance is increasing year by year. Since the advent of cisplatin (CDDP) in the 1970s, various cytotoxic anti-cancer agents have been indicated for HNSCC. Following the results of the Bonner study for locally advanced HNSCC (LA-HNSCC) in 2006 and the EXTREME study for recurrent/metastatic HNSCC (R/M-HNSCC) in 2008, cetuximab was approved as the first molecular-targeted drug for HNSCC in the world, and it has come to be used in combination with radiotherapy or cytotoxic anti-cancer drugs [1,2]. After that, ICIs were approved for various solid tumors, and nivolumab was approved as a second-line treatment for platinum-resistant R/M-HNSCC in 2017 (CheckMate 141 study) [3]. In 2019, pembrolizumab monotherapy and cytotoxic anti-cancer drug combination therapy was approved as the first-line treatment for R/M-HNSCC and LA-HNSCC (KEYNOTE-048 study) [4]. Furthermore, in recent years, “Oncogene panel tests” have also been actively used to search for subsequent treatments after these second-line or third-line treatments.

Here we conducted a literature review on clinical trials in molecular targeted therapy in HNSCC. Based on previous literatures, we show the current trends and future prospects related to their biomarkers, which have made remarkable progression in recent years, including the topic of photoimmunotherapy, which was most recently approved for HNSCC in Japan.

## 2. Anti-Cancer Drug Therapy in HNSCC

### 2.1. The History of Cytotoxic Anti-Cancer Drugs

With the advent of CDDP in the 1970s, chemotherapy for HNSCC greatly progressed. The combination of FP therapy (CDDP + 5-fluorouracil (5-FU)) is still positioned as a standard regimen for HNSCC, beginning after a report from Kish et al. in 1982 [5]. Kish et al. showed that the response rate (complete response (CR) + partial response (PR)) is 88.5% for untreated patients [5]. CDDP and carboplatin (CBDCA) have been widely applied as platinum-based anti-cancer drugs from 2000, and since then, taxane-based anti-cancer drugs, such as docetaxel (DTX) and paclitaxel (PTX), have been introduced for HNSCC from 2010. The response rate of the combination therapy of DTX and CDDP in LA-HNSCC was 33–53%, which were good results [6,7,8]. Table 1 shows the main variations of anti-cancer drugs (including cetuximab, nivolumab and pembrolizumab) in HNSCC.

### 2.2. Current Trends of Anti-Cancer Drug Therapy

The combination of CDDP and radiotherapy (chemo-radiotherapy: CRT) is used as the standard treatment for LA-HNSCC. With the advent of cetuximab, the combination of cetuximab and radiotherapy (bio-radiotherapy: BRT) may be a treatment option, but at present there is no trial that proves that BRT is non-inferior to CRT. In 2016, Magrini et al. reported that CRT tended to be superior to BRT in local control rate and overall survival [9]. In addition, there are no reports that BRT is superior to CRT in terms of safety associated with adverse events.

Figure 1 shows the treatment algorithm (combination of molecular targeted and cytotoxic drugs) for R/M-HNSCC, which take into account new findings in recent years [10,11,12]. First, if the performance status is relatively good and it is judged that the cases can be treated with anti-cancer drugs, they are roughly classified into platinum-resistant cases (recurrence within 6 months) or chemotherapy untreated cases [3]. Furthermore, the latter is subdivided using the combined positive score (CPS) [4]. Although there are various arms, it is common that cetuximab and ICIs are used alternately for each treatment line. For the evidence of the algorithm, please refer to the description of clinical trials for cetuximab, nivolumab and pembrolizumab.

## 3. Cetuximab

### 3.1. The History of Cetuximab

Cetuximab is a human/mouse chimeric monoclonal antibody targeting epidermal growth factor receptor (EGFR), first approved in the world. It is used for the treatment of metastatic colon cancer, metastatic non-small cell lung cancer, and HNSCC. For colon cancer, cetuximab was first approved in Switzerland, USA, and EU from 2003 to 2004 as a second-line treatment for EGFR-positive unresectable advanced or recurrent colorectal cancer. In 2008, it was approved as a first-line treatment for EGFR-positive and KRAS wild-type colorectal cancer. On the other hand, for HNSCC, based on the results of the Bonner study in 2006, “cetuximab + RT” and “cetuximab single agent use after BRT” have been approved for LA-HNSCC [1]. In 2008, the results of the EXTREME study were published. For R/M-HNSCC, “Cetuximab + CDDP (or CBDCA) + 5-FU” was approved in many countries and was introduced to Japan in 2012 [2].

### 3.2. Current Trends of Cetuximab in LA-HNSCC and R/M-HNSCC 

Cetuximab has been proven to be useful in various clinical trials in combination with radiotherapy and chemotherapy such as CDDP. The Bonner study compared 424 LA-HNSCC patients with two groups, the combination group (radiotherapy plus cetuximab) and radio-monotherapy group [1]. As a result, the median local disease control period was 24.4 months in the combination group and 14.9 months in the radio-monotherapy group (*p* = 0.005), and the median overall survival (OS) was 49.0 months and 29.3 months, respectively (*p* = 0.03). Therefore, the cetuximab combination group was significantly superior in local control and survival.

The EXTREME study compared 442 R/M-HNSCC patients with two groups, the FP alone group (CDDP/CBDCA + 5-FU) and the combination group (FP plus cetuximab) [2]. As a result, the median OS was 10.1 months in the combination group versus 7.4 months in the FP alone group (*p* = 0.04), and the median progression-free survival was 5.6 months and 3.3 months, respectively (*p* = 0.001). Thus, a significantly prolonged survival was shown in the cetuximab combination group.

In addition to the EXTREME study, clinical trials for R/M-HNSCC include the GORTEC 2008-03 trial (CDDP + DTX + cetuximab) [13], Hitt trial (Weekly PTX + cetuximab) [14], CSPOR-HN02 trial (PCE regimen; PTX + CBDCA + cetuximab) [15], and any other widely applied regimens.

### 3.3. Future Prospects of Cetuximab in HNSCC

High EGFR expression is said to be found in 25–77% of colon cancers and 90% or more of HNSCC [16,17], when ligands such as EGF and transforming growth factor-α (TGF-α) bind to EGFR, they form a dimer with EGFR or other human epidermal growth factor receptor (HER) family members. Therefore, autophosphorylation of the intracellular tyrosine kinase domain, and activating further downstream Ras-Raf-MAPK and PI3K-Akt pathways are deeply involved in cancer growth and metastasis.

Initially, cetuximab was used exclusively in colon cancer cases in which EGFR expression was observed in tumor cells by immunostaining, but subsequent studies showed good response rates in colon cancer cases in which EGFR expression was negative [18]. Therefore, it was clarified that the intensity of EGFR expression did not correlate with the therapeutic effect of cetuximab [19]. Currently, EGFR immunostaining is not recommended for determining the indication of anti-EGFR antibody [20]. Anti-EGFR antibody therapy for unresectable colon cancer was found to be ineffective in cases with RAS (KRAS/NRAS) gene mutations (about 40% of colorectal cancers), and RAS genetic testing was established as a companion diagnostic tool [21]. Recently, other genetic abnormalities, such as gene mutation of BRAF [22], PIK3CA [23], and EGFR extracellular domain (ECD) [24], gene amplification of HER2 [25] and mesenchymal-epithelial transition factor (MET) [26], have been reported as acquisition resistance factors of anti-EGFR antibody [27].

Although a biomarker for predicting the therapeutic effect of cetuximab has not been established in HNSCC, it has been suggested to be associated with rash, which is a typical adverse event of cetuximab. In an additional report from the Bonner study, OS in BRT patients was 25.6 months in the group with Grade 1 or lower rash, whereas it was 68.8 months in the group with Grade 2 or higher [28], indicating that the OS of the severe rash group was significantly superior. Furthermore, in the report by Saltz L et al., the response rate and disease control rate of the cases with skin disorders, such as rash, tended to be higher than the cases without rash, when cetuximab was applied to R/M-HNSCC [29]. Rash, which is a typical adverse event of cetuximab, is the most clinically easy-to-understand index and is attracting attention as a clue for the development of biomarkers in the future.

## 4. Immune Checkpoint Inhibitors (Nivolumab and Pembrolizumab)

### 4.1. The History of ICIs

HNSCC is considered to be a malignant tumor in which the immune surveillance mechanism is suppressed, and the rationale for this is a decrease in the absolute number of lymphocytes, a decrease in NK cell function, a decrease in the antigen presenting function of antigen presenting cells, tumor infiltrating lymphocyte function decline, regulatory T cell function enhancement, and avoidance from T cell immunity due to persistent viral infection [30]. These facts are the rationale for the promising ICIs in HNSCC. Smoking, alcohol consumption, mechanical irritation, etc. are strongly associated with the onset of HNSCC. In particular, oral cancer is often carcinogenic due to the above-mentioned continuous and long-term external stimuli. Therefore, it is considered to have a high tumor mutational burden on somatic cells [31]. A correlation between the amount of this gene mutation and the effect of ICIs has already been reported in malignant melanoma and lung cancer [32,33], and a strong effect is expected in oral cancer. Furthermore, tumor cells of HNSCC have a relatively high expression rate of PD-L1 (programmed cell death 1-ligand 1) and are in an immune-avoidance state. In addition, PD-L1 expression rates are even higher in HNSCCs associated with persistent viral infections, such as HPV (human papilloma virus) and EBV (Epstein–Barr virus) [34,35]. From these reasons, ICIs are considered to be a good therapeutic modality for HNSCC.

### 4.2. Current Trends of ICIs in R/M-HNSCC

Currently, the ICIs used clinically for R/M-HNSCC are the anti-PD-1 antibodies, nivolumab and pembrolizumab. Evidence of nivolumab is the CheckMate-141 trial. The subject was HNSCC with recurrence/metastasis, which recurred/aggravated within 6 months after treatment with a platinum-containing regimen. In a randomized phase III trial, the primary endpoint, OS, was significantly prolonged compared to chemotherapy with the control arm of methotrexate, docetaxel, or cetuximab (Nivolumab group: 7.7 months, standard treatment group: 5.1 months, hazard ratio: 0.70). Furthermore, it was shown that the Nivolumab group was significantly better in terms of symptom relief and quality of life (QOL) maintenance [3,36]. These results make nivolumab a new treatment option for platinum-resistant R/M-HNSCC. We think that comprehensive treatment selection such as concomitant use with other anti-cancer agents may be necessary. Prior to the start of this study, tissue biopsy was performed to confirm PD-L1 expression in tumor cells. However, PD-L1 expression was not correlated with the response of nivolumab. Finding a companion diagnostic tool will be required.

Next, the evidence for pembrolizumab is the KEYNOTE-048 study, which examined the expression level of PD-L1 in tumor tissues and showed that it is effective to use other anti-cancer agents in combination depending on the amount of PD-L1 expression [4]. For the expression level of PD-L1, the combined positive score (CPS), which is the ratio of the total PD-L1-positive tumor cells and PD-L1-positive immune cells in all tumor tissue cells, was adopted (Figure 2) [37]. This study compared “FP/FC (5-FU + CDDP/5-FU + CBDCA) + pembrolizumab combination therapy” with “FP/FC + cetuximab combination therapy”, which is the standard initial treatment for platinum-sensitive R/M-HNSCC. The results showed prolongation in OS in cases with CPS ≥ 20% and CPS ≥ 1%. In addition, pembrolizumab monotherapy showed prolongation in OS in cases with CPS ≥ 20% and CPS ≥ 1% and proved non-inferiority in the entire population. This result suggests the usefulness of CPS as a biomarker for predicting prognosis when considering administration of pembrolizumab. Based on this, pembrolizumab has been added as an initial treatment option for R/M-HNSCC, which has expanded the treatment options. However, the CPS introduced by the founder of KEYNOTE-048 is a subjective evaluation of trained pathologists, and it cannot be said that there is a universal index yet. In the future, it is hoped that evaluations will be standardized using computer-based image analysis tools or similar.

When using ICIs, it is necessary to pay attention to immune-related adverse events (irAEs), which are excessive reactions to normal cells associated with immune activation. Typical affected organs are the digestive tract, lungs, liver, skin, and endocrine organs, including the thyroid gland and adrenal glands. Although the onset of irAEs in these organs varies, it often develops at a median of between 5 and 15 weeks [38]. Various guidelines can be used as a reference for details on how to deal with each organ [39]. It is very important not to forget regular examinations and to take prompt action, such as cooperating with specialized clinical departments when necessary.

### 4.3. Future Prospects of ICIs in HNSCC

In R/M-HNSCC, we have entered an era in which ICIs are used from the first-line to the second-line treatment (platinum resistance). In recent years, if cytotoxic anti-cancer drug treatment, such as platinum or 5-FU, is used again after using an ICI as a second-line treatment, the immune environment in the tumor tissue changes and the drug sensitivity becomes higher. Its usefulness has also been reported in salvage chemotherapy, which improves sensitivity to tumors [40,41,42]. As a biomarker for the response of salvage chemotherapy, it has been reported that the lower the neutrophil-to-lymphocyte ratio (NLR) and CRP before chemotherapy, the higher the response rate [43]. So far, it has been reported that systemic inflammatory responses such as fever (tumor fever) and weight loss may occur in many carcinomas, especially in advanced cases. It has also been shown to have a poor prognosis for patients with a strong inflammatory response [44,45,46]. NLR and CRP indicate the strength of the inflammatory response, and these markers may also be useful in predicting prognosis in head and neck cancer. In particular, the administration of ICIs changes the micro-immune environment of the tumor, so examining the inflammatory response after administration of ICIs may be a predictor of the effectiveness of salvage chemotherapy. Although more cases still need to be accumulated, NLR and CRP may be useful as prognostic biomarkers. In addition, as a characteristic of immunotherapy, the response rate on advanced cancer is relatively low, but in cases of stable disease (SD), the tumor suppression effect is long-lasting (durable response) due to the mechanism mediated by the host immune system, and prolonging survival is achieved. Superior potential is also shown in the 2-year survival data from the CheckMate-141 trial [47]. In other words, there is a possibility that treatment can be continued for a long period of time while maintaining a certain level of QOL. Furthermore, in recent years, liquid biopsy tests have attracted attention for detecting microsatellite instability and tumor mutational burden. The liquid biopsy tests may be used as a predictor of the response of ICIs [48].

Thus, ICI therapy is a very attractive treatment option and will continue to provide significant benefits to patients with HNSCC. In the future, after predicting the response by liquid biopsy etc., we will select an appropriate ICI, and use it in combination with a molecular target drug such as a VEGF (vascular endothelial growth factor) inhibitor, or anti-PD-L1, which has already been approved for other cancers such as melanoma. Clinical applications including combined use of an anti-PD-1 antibody drug and anti-cytotoxic T-lymphocyte antigen-4 (anti-CTLA-4) antibody drug is expected.

## 5. Multi-Oncogene Panel Tests

The oncogene panel test is a “gene profiling test” by using next generation sequencing (NGS) to examine the changes in hundreds of cancer-related genes at the same time [49]. Currently, two types of the gene panel test for cancer, “OncoGuide^TM^NCC Oncopanel System” and “FoundationOne^®^CD_X_ Oncogenome Profile” are mainly used [50,51,52]. Both of them are analyzed in formalin fixed tumor specimens by using NGS. Unstained specimens embedded in paraffin are used to obtain the information that can be supposed as a reference for treatment, such as a selection of therapeutic agents, prediction of prognosis, and diagnosis of cancer type. In the former gene panel test, 114 genes can be analyzed, and in the latter, 324 genes can be analyzed. Patients who are subject to the genetic panel test are (i) solid cancer patients who have not had standard treatment, or (ii) solid cancer patients who have local progression or metastasis and have completed standard treatment. The former mainly targets cancers for which standard treatments are scarce, such as rare cancers and childhood cancers, and the latter covers many other cancer types. That is, most of the target patients have advanced or recurrent cancer, and tests are performed to find a treatment that follows standard treatment. The most promising gene panel test for cancer is the selection of therapeutic agents associated with genetic alterations. Patients who have not had or have completed standard treatment are subject to a gene panel test for cancer, so the administration of therapeutic agents is mainly off-label use of approved drugs or administration of molecular targeted drugs in clinical trials. Therefore, the biggest problem in gene panel tests for cancer is that the administration of the therapeutic drug is mainly off-label or clinical trial use even if a genetic change related to a therapeutic drug is detected, so the proportion of patients who are actually administered is limited [49]. In a prospective study called the TOP-GEAR project in the NCC oncopanel test, 112 of the 187 patients tested were positive for genetic changes that could be expected to be effective with molecular targeted drugs, but 25 patients (15%) received treatment, such as off-label use of approved drugs, commensurate with genetic changes [50]. In the MSK-IMPACT trial, this proportion was also 11% [53]. In other words, the biggest problem of the gene panel test is that even if genetic changes are detected, few patients are eligible for drug treatment, because the drug is absent or unavailable under development. These clinical trials have shown that the proportion of the drug is as low as about 10% of all test cases. Basic research is also actively progressed to create indicators for the administration of known molecular targeted drugs such as EGFR and HER2 [54]. Unfortunately, there are many genes for which there are no molecular targeted drugs associated with them, although there are changes in various cancers. The active mutation of KRAS is a typical example, and drugs that specifically bind to the KRAS mutant protein have been developed, and clinical trials will be shown as good therapeutic effects [55]. Therefore, it is expected that the proportion of patients linked to drug treatment will increase as the number of molecular targeted therapeutic agents associated with gene changes. 

In recent years, the single-cell RNA-sequence (scRNA-seq) reveals that the diverse malignant, stromal, and immune cells in tumors affect growth, metastasis, and response/resistance to therapy. Malignant tumor cells vary within and between tumors in their expression of signatures related to cell cycle, stress, hypoxia, epithelial differentiation, and partial epithelial-to-mesenchymal transition (p-EMT) [56]. In various cases, understanding intra-tumoral expression heterogeneity in epithelial tumors is very important, but it is very difficult. At present, gene panel tests do not cover these heterogeneities among epithelial cells, and further development is required in the future. The precision and personalized medicine by the target therapy using a combination of several drugs based on the heterogeneity of tumors may be used in the near future.

With regard to oral cancer, we may encounter locally advanced cases, standard treatment-resistant recurrence/metastasis cases, or cancer types for which no treatment algorithm has been established. Cisplatin is the main therapeutic agent for HNSCC. Currently, cetuximab, which is an anti-EGFR antibody, and nivolumab and pembrolizumab, which are anti-PD-1 antibodies, have been approved and treated as molecular targeted therapeutic agents that can be used for oral cancer [57]. In addition, clinical trials of molecular targeted therapeutic agents such as ipilimumab, which is an anti-CTLA-4 antibody [58], and avelumab and atezolizumab, which are anti-PD-L1 antibodies [59], have been performed. However, which agent should be used and how much effect can be expected differ from each patient when using these molecular targeted therapeutic agents, and it is considered that there is no response to these agents at all. In addition, it is supposed that off-label drugs may be effective. Therefore, it is important to confirm which gene is expressed to what extent by a gene panel test in order to perform appropriate treatment even in the field of HNSCC. However, gene panel tests are not widely used in the field of HNSCC at this moment. In the future, many gene panel tests will be performed in the field of HNSCC, and it is expected that it will help in the selection of appropriate treatment methods.

## 6. Photoimmunotherapy in HNSCC

In September 2020, the Japanese government approved cetuximab saratolacan (previously known as RM-1929, trade name: Akalux) as the treatment product of near infrared photoimmunotherapy (NIR-PIT) for unresectable locally advanced or recurrent HNSCC [60]. NIR-PIT was developed by Dr. Hisataka Kobayashi, who is a senior investigator in the National Cancer Institute (NCI) / National Institutes of Health (NIH), as a treatment method that kills only cancer cells and causes almost no damage to normal cells and has been attracting attention in recent years [61,62,63]. Cetuximab saratolacan is a chemical conjugate of the photosensitizer IRDye700DX (IR700) with cetuximab, which targets EGFR, and was developed by Rakuten Medical Japan. The treatment consists of the intravenous injection of cetuximab saratolacan, which binds to HNSCC cells with high levels of EGFR expression, followed by illumination of the tumor with NIR light (690 nm) for photodynamic therapy (Figure 3). The medical laser device “BioBlade Laser System” that emits the NIR light was also approved for manufacture and distribution at the same time. NIR-PIT using Akalux is now transitioning to fast-track global Phase III clinical trial in locally advanced or recurrent HNSCC patients, who are resistant to multiple conventional treatments [64,65]. When exposed to NIR light, Akalux induces a highly selective and rapid necrotic/immunogenic cell death (ICD) only in target-positive cells. The ICD occurs as early as 1 min after exposure to NIR light, but immediately adjacent target-negative cells are unharmed [66]. ICD induced by NIR-PIT promoted maturation of immature dendritic cells (DCs) to maturated DCs that primed cytotoxic T-cells to react with cancer-related antigens released from destroyed cancer cells [66]. NIR-PIT causes highly selective necrotic/ICD only in cancer cells, thus triggering a potent anti-cancer immune response, so it is expected to develop from the head and neck region where functional preservation and aesthetics of post-treatments are required.

The international multicenter clinical trial of NIR-PIT using Akalux in locally advanced or recurrent HNSCC began in 2015, and the results of the Phase IIa trial were reported at ASCO (American society of clinical oncology) in 2019 [67]. Thirty patients with locally advanced or recurrent HNSCC for resistance of multiple conventional treatments were enrolled, and as a result, the objective response rate was 50% (15/30 cases), of which CR was 16.7% (5/30 cases) and PR was 33.3% (10/30 cases), and the disease control rate, including 11 cases of SD, was 86.7% (26/30 cases), which was very good [67]. In addition, most of the reported adverse events were mild to moderate in severity, with 96.7% (29/30 cases) of Grade 1 patients and 83.3% (25/30 cases) of Grade 2 patients [67]. From these results, NIR-PIT using Akalux in locally advanced or recurrent HNSCC is a wonderful treatment that has relatively mild treatment-related adverse events and is expected to be more effective than existing cetuximab or ICI (nivolumab and pembrolizumab). Photoimmunotherapy with cetuximab salatracan is a useful treatment option that can be widely applied to various HNSCCs, not limited to oral and oropharyngeal cancer, which are EGFR-positive tumors in anatomical areas that are easily exposed to near-infrared light. It is also possible to irradiate with an endoscope, and I think it can be applied to both outpatients and inpatients. 

## 7. Epigenetic-Targeted Therapy in HNSCC

The importance of drug epigenetic alternations concern heritable yet reversible changes in histone or DNA modifications that regulate gene activity beyond the underlying sequence, and epigenetic dysregulation is often related to human diseases, especially cancer [68]. Currently, antibody drugs such as cetuximab, nivolumab and pembrolizumab are used as molecular targeted therapies for HNSCC, but clinical trials of epigenetic-targeted therapy that has already been clinically applied to hematological malignancies (malignant lymphoma, multiple myeloma, etc.) are also ongoing for HNSCC [69]. A histone deacetylase (HDAC) inhibitor is a typical drug among drugs targeted at epigenetic enzymes. The HDAC inhibitor “romidepsin” was used in a clinical phase II trial for HNSCC patients, and it was reported that the single agent romidepsin has limited activity for the treatment of HNSCC but can effectively achieve tumor-associated HDAC inhibition [70]. 

## 8. Conclusions

The importance of drug therapy for HNSCC is growing year by year. We conducted a literature review on clinical trials of molecular targeted therapy including photoimmunotherapy and the biomarker in HNSCC. This review shows current trends and future prospects. It is expected that the algorithm for HNSCC drug therapy will be further subdivided in the future, leading to improvements in survival rate.

## Figures and Tables

**Figure 1 ijms-22-00240-f001:**
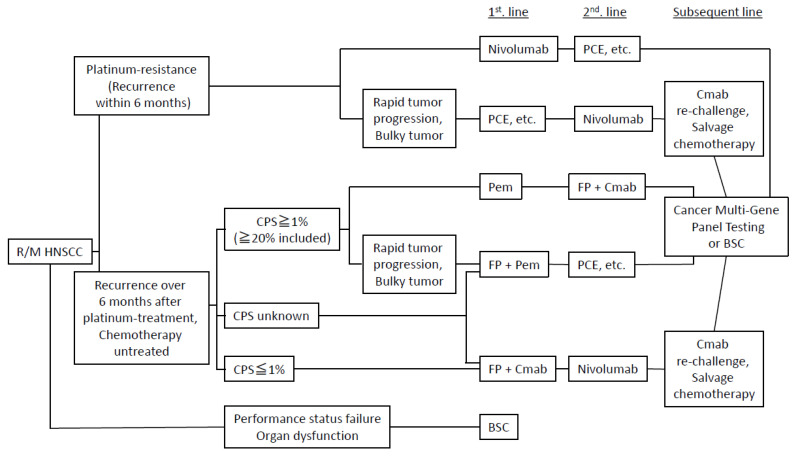
Treatment algorithm of R/M HNSCC. FP: 5-FU + cisplatin, PCE: paclitaxel + carboplatin + cetuximab, Cmab: cetuximab, Pem: pembrolizumab, CPS: combined positive score, BSC: best supportive care. Citations and modifications of *Jpn J Chemother (Gan To Kagaku Ryoho)*. **2020**, 47, 1050–1054.

**Figure 2 ijms-22-00240-f002:**
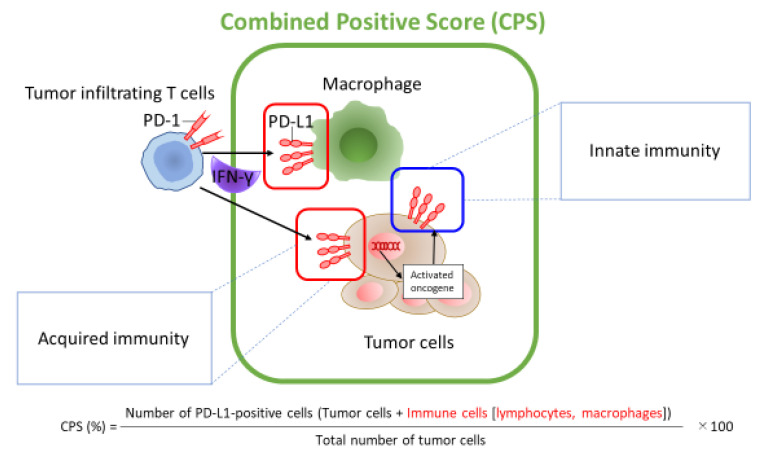
Mechanisms for PD-L1 expression and definition of combined positive score (CPS). Citations and modifications of *Nat Rev Cancer.*
**2016**, 16, 275–287. Innate expression of membranous PD-L1 by tumor cells is thought to be driven by dysregulated signaling pathways, or chromosomal alterations and amplifications. In contrast, adaptive focal expression of PD-L1 by tumor cells and macrophages occurs at the interface of tumor cell nests with immune infiltrates secreting pro-inflammatory factors such as interferon-γ. The ligation of PD-L1 with PD-1 molecules will down-modulate T cell function, essentially creating a negative feedback loop that dampens antitumor immunity [37].

**Figure 3 ijms-22-00240-f003:**
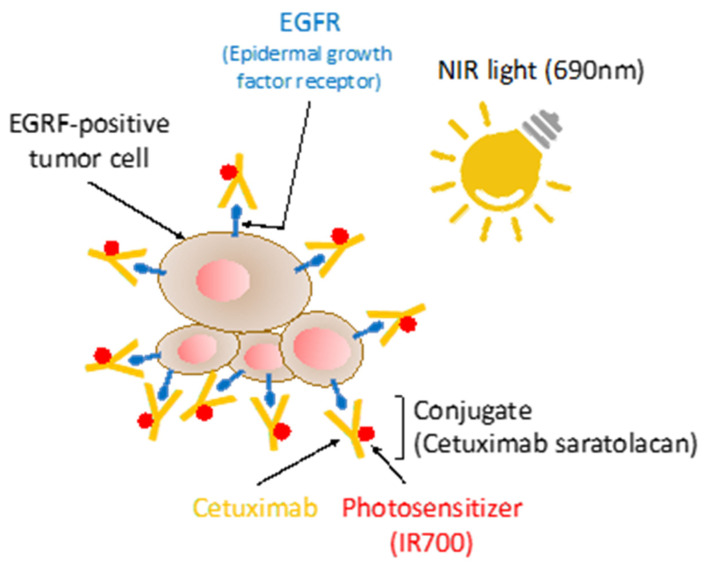
Near infrared photoimmunotherapy (NIR-PIT). Cetuximab-bound IR700 selectively binds to the surface of EGFR-positive tumor cells. The photo-activation at 690 nm selectively kills EGFR-expressing cells, thus allowing for targeted photodynamic therapy and the induction of immunogenic cell death. This triggers a systemic immune response that contributes to the eradication of malignant cells [60].

**Table 1 ijms-22-00240-t001:** Variation of anti-cancer drugs in HNSCC.

Category	Sub-Category	Drug Name	Mechanism	Main Site of Drug Metabolism	Main AEs	The Others of AEs	Clinical Research	Effectiveness
Cytotoxic anti-cancer agents	Platinum-based	Cisplatin	DNA crosslink	Kidney	Bone-marrow suppression, kidney dysfunction	Allergy, peripheral nerve disorder, nausea	Ref. [5]	RR: 88.5%
		Carboplatin	Cell-cycle non-specific					
	Taxane-based	Docetaxel	Depolymerization inhibitor	Liver	Bone-marrow suppression, hair loss, nausea	Allergy, edema, peripheral nerve disorder	Ref. [6,7,8]	RR: 33–53%
		Paclitaxel	Stop at M phase					
	Pyrimidine-based	5-fluorouracil (5-FU)	DNA synthesis inhibitor	Liver	Bone-marrow suppression, mucositis, nausea	Allergy, Myocardial ischemia, Diarrhea	Ref. [5]	RR: 88.5% (Platinum-combination therapy)
		Tegafur, Gimeracil, and Oteracil potassium (S-1)	S phase specific					
Molecular-targeted agents	Antibody-drug	Cetuximab	EGFR inhibitor	-	Rash, skin dryness, paronychia	Infusion reaction, hypomagnesemia, interstitial pneumonia	LA-HNSCC: Ref. [1]	OS: 49 months
							R/M-HNSCC: Ref. [2]	OS: 10.1 months
Immune checkpoint inhbitor	Anti PD-1 antibody	Nivolumab	PD-1 receptor inhibitior	-	Diarrhea, Dysthyroidism, Rash	Colitis, diabetes mellitus, interstitial pneumonia	Platinum-resistance: Ref. [3]	OS: 7.7 months
		Pembrolizumab					Ref. [4]	OS: 14.7/14.9 months (Pem-combination/monotherapy)

EGFR: epidermal growth factor receptor, PD-1: programmed cell death-1, AEs: adverse events, Pem: pembrolizumab, OS: median over all survival, RR: response rate, LA-HNSCC: locally advanced head and neck squamous cell carcinoma, R/M-HNSCC: recurrent/metastatic head and neck squamous cell carcinoma.

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
