# Peer review of "Current Trends and Future Prospects of Molecular Targeted Therapy in Head and Neck Squamous Cell Carcinoma"

_ijms, 2020, doi:10.3390/ijms22010240_

Round 1

Reviewer 1 Report

  1. In this manuscript, there is no any discussion about epigenetic therapy. There were clinic trials in head and neck cancer with epigenetic therapy . The author should discuss about the epigenetic therapy.

Condition: Design: Sample size: Phase: Current status: NCT

Squamous cell carcinoma of the head and neck: Romidepsin: 14: II: Completed: NCT00084682

Metastatic thyroid cancer, Panobinostat, 13, II, Completed, NCT01013597

Reference: Cheng, Y., He, C., Wang, M. et al. Targeting epigenetic regulators for cancer therapy: mechanisms and advances in clinical trials. Sig Transduct Target Ther 4, 62 (2019). https://doi.org/10.1038/s41392-019-0095-0

  1. In the section of multi-oncogene panel tests, the authors discussed with

" In other words, the biggest problem of the gene panel test is that even if genetic changes are detected, few patients are eligible for drug treatment, because the drug is absent or unavailable under development."

However, by the advance of single cell analysis, cancer heterogeneity was explored. The authors should discussed about how the impacts will be by single cell analysis compare the convention multi-oncogene panel test.

Reference: Sidharth V. Puram, Itay Tirosh, Anuraag S. Parikh, Anoop P. Patel, Keren Yizhak, Shawn Gillespie, Christopher Rodman, Christina L. Luo, Edmund A. Mroz, Kevin S. Emerick, Daniel G. Deschler, Mark A. Varvares, Ravi Mylvaganam, Orit Rozenblatt-Rosen, James W. Rocco, William C. Faquin, Derrick T. Lin, Aviv Regev, Bradley E. Bernstein, Single-Cell Transcriptomic Analysis of Primary and Metastatic Tumor Ecosystems in Head and Neck Cancer, Cell, Volume 171, Issue 7, 2017, Pages 1611-1624.e24.

Author Response

We appreciate that the referee is enthusiastic about certain aspects. He/she asks that we address the following specific issues.

1.      In this manuscript, there is no any discussion about epigenetic therapy. There were clinic trials in head and neck cancer with epigenetic therapy. The author should discuss about the epigenetic therapy.

As suggested by this reviewer, we added the topic, “7. Epigenetic-targeted therapy in HNSCC” as the following:

Epigenetic-targeted therapy in HNSCC

The importance of drug Epigenetic alternations concern heritable yet reversible changes in histone or DNA modifications that regulate gene activity beyond the underlying sequence, and epigenetic dysregulation is often related to human diseases, especially cancer [Sig Transduct Target Ther. 2019, 4, 62.]. Currently, antibody drugs such as cetuximab, nivolumab and pembrolizumab are used as molecular targeted therapies for HNSCC, but clinical trials of epigenetic-targeted therapy that has already been clinically applied to hematological malignancies (malignant lymphoma and multiple myeloma, etc.) are also ongoing for HNSCC [Int. J. Mol. Sci. 2017, 18, 1506.]. Histone deacetylase (HDAC) inhibitor is a typical drug among drugs targeted at epigenetic enzymes. The HDAC inhibitor “Romidepsin” was used as the clinical phase II trial for HNSCC patients, and it was reported that single agent romidepsin has limited activity for the treatment of HNSCC but can effectively achieve tumor-associated HDAC inhibition [Oral Oncol. 2019, 48, 1281-1288.].

2.      In the section of multi-oncogene panel tests, the authors discussed with " In other words, the biggest problem of the gene panel test is that even if genetic changes are detected, few patients are eligible for drug treatment, because the drug is absent or unavailable under development." However, by the advance of single cell analysis, cancer heterogeneity was explored. The authors should discussed about how the impacts will be by single cell analysis compare the convention multi-oncogene panel test.

Thank you for a valuable comment. We added the topic "Effect of single cell analysis on multi-oncogene panel test" as the following:

In recent years, the single-cell RNA-sequence (scRNA-seq) reveals that the diverse malignant, stromal, and immune cells in tumors affect growth, metastasis, and response/resistance to therapy. Malignant tumor cells vary within and between tumors in their expression of signatures related to cell cycle, stress, hypoxia, epithelial differentiation, and partial epithelial-to-mesenchymal transition (p-EMT) [Cell. 2017, 171, 7, 2017, 1611-1624.]. In various cases, understanding intra-tumoral expression heterogeneity in epithelial tumors is very important, but it is very difficult. At the present, gene panel tests do not cover these heterogeneities among epithelial cells, and further development is required in the future. The precision and personalized medicine by the target therapy using a combination of several drugs based on the heterogeneity of tumors may be used in the near future.

Reviewer 2 Report

Dear Authors,

The present article is well conceptualized, structured and easy to read. This work is classic review. Not systematic review. There are some points that need improvement. I suggest major changes:

1) First of all, for the possibility to address certain words and sentences, the authors should fix the numeration of lines and pages.

2) Abstract: Furthermore, in September 2020, cetuximab saratolacan for photoimmunotherapy has been approved for HNSCC in Japan. - please delete this sentence.

3) Figure 1 is missing. I cannot judge.

4) Rewrite article. Write materials and methods according to the criteria of the PRISMA guidelines for systematic review.

5) Data from the text on the research and effectiveness of individual drugs in the HNSCC, please include in the table.

6) The names of the genes should be written in italics.

7) Wrong numbering in the article: 3. Immune checkpoint inhibitors (Nivolumab and Pembrolizumab) should have number four

4. Multi-oncogene panel tests should have number five

5. Photoimmunotherapy in HNSCC should have number six

6. Conclusions should have number seven

8) The article should be shortened. When quoting authors, make only important finding.

9) Authors should specify all the Abbreviations used in the text and, if possible, insert the abbreviations list after the Conclusions.

Author Response

This Reviewer is enthusiastic about our results stating that: “The present article is well conceptualized, structured and easy to read”. He/she asks that we address the following specific issues.

  1. First of all, for the possibility to address certain words and sentences, the authors should fix the numeration of lines and pages.

IJMS provides us the format of the manuscript. Therefore, it is difficult to fix the numeration of lines and pages. We are sorry that we could not meet your request.

  1. Abstract: Furthermore, in September 2020, cetuximab saratolacan for photoimmunotherapy has been approved for HNSCC in Japan. - please delete this sentence.

As suggested by this reviewer, we deleted this sentence.

  1. Figure 1 is missing. I cannot judge.

When the submitted “Word file” was converted to a PDF file, Figure 1 may be missing. We will submit a PDF version of our revised manuscript. We are sorry for bothering you.

  1. Rewrite article. Write materials and methods according to the criteria of the PRISMA guidelines for systematic review.

This review article is not a systematic review. Therefore, materials and methods cannot be described.

  1. Data from the text on the research and effectiveness of individual drugs in the HNSCC, please include in the table.

As suggested by this reviewer, we included "the research and effectiveness of individual drugs in the HNSCC" in Table 1.

  1. The names of the genes should be written in italics.

As pointed out by this reviewer, we have changed the genes names to italic.

  1. Wrong numbering in the article:
  2. Immune checkpoint inhibitors (Nivolumab and Pembrolizumab) should have number four
  3. Multi-oncogene panel tests should have number five
  4. Photoimmunotherapy in HNSCC should have number six
  5. Conclusions should have number seven

Thank you very much for pointing out. We have fixed all the above numbering.

  1. The article should be shortened. When quoting authors, make only important finding.

As suggested by this reviewer, we have shortened the citation of the clinical trial’s results of photoimmunotherapy in the latter half of page 8.

  1. Authors should specify all the Abbreviations used in the text and, if possible, insert the abbreviations list after the Conclusions.

As your suggestion, we provided the abbreviations list after “Conclusions”.

Reviewer 3 Report

In the submitted review article authors conducted a literature review on clinical trials in molecular targeted therapy in HNSCC.

The submitted review is a comprehensive article of available molecular target based therapy in head and neck cancer. Its main highlight is the cetuximab combined photosensitizer NIR laser therapy, which might be very promising. Since the available treatment options including the KEYNOTE-048 study do not perform with high satisfaction, more critical discussion could be expected by the authors.

Also the CPS introduced by the founders of the KEYNOTE-048 is a subjective evaluation of trained pathologists, which lacks all independent computer-based image analysis tools available in the 21st. century. Moreover, the first chapter on page 5 is not understandable at all. What is superiority, why CPS is useful as biomarker, biomarker for what? predictive for what kind of outcome?, what was compared with what? treatment with 5-FU, Cisplatin, cetuximab and pembrolizumab all inclusive? Please provide clearer description and explain everything in enough detail, that we can understand. Please see in your review article the results critically and do not only repeat the dubious results of the original studies.

A further point: several useful and important facts are included in the review article, which could be discussed more in detail, as for example: the lower the neutrophil-to-lymphocyte ratio (NLR) and CRP before chemotherapy, the higher the response rate. The achive this aim the influence of immune checkpoint is definitely important. Do the authors see any other option here, especially from the field of treatments for the chronic inflammations?

The chapter multi-oncogene panel tests is also interesting, containing up to 324 genes utilized, and finally authors end up still with CTLA-4 and PD-L1 antibodies which are currently used, and are nothing new. If it is like this, also critical remarks could be expected here.

An absolute highlight in the submitted review article is the report about Cetuximab saratolacan, which has a tumor specific potential and is absolutely novel approach. Please provide more details on this especially if it is only usable for oral cancer or also oropharynx, etc. could it be also used during pan-endoscopy or only in ambulant setting?

Minor Detailed Comments:

Figure 1 is missing from the submission. Please include it.

Author Response

We appreciate that the referee is enthusiastic about certain aspects. He/she asks that we address the following specific issues.

  1. CPS introduced by the founders of the KEYNOTE-048 is a subjective evaluation of trained pathologists, which lacks all independent computer-based image analysis tools available in the 21st. century. Moreover, the first chapter on page 5 is not understandable at all. What is superiority, why CPS is useful as biomarker, biomarker for what? predictive for what kind of outcome?, what was compared with what? treatment with 5-FU, Cisplatin, cetuximab and pembrolizumab all inclusive?

Thank you for a valuable comment. We have revised the chapter about "CPS" that you pointed out as the following:

Next, the evidence for pembrolizumab is the KEYNOTE-048 study, which examined the expression level of PD-L1 in tumor tissues and showed that it is effective to use other anti-cancer agents in combination depending on the amount of PD-L1 expression [4]. For the expression level of PD-L1, the Combined Positive Score (CPS) that is the ratio of the total of PD-L1-positive tumor cells and PD-L1-positive immune cells in all tumor tissue cells was adopted (Figure 2) [34]. This study compared “FP/FC (5-FU + CDDP/5-FU + CBDCA) + pembrolizumab combination therapy” with “FP/FC + cetuximab combination therapy”, which is the standard initial treatment for platinum-sensitive R/M-HNSCC. The results showed prolongation in OS in cases with CPS≥20% and CPS≥1%. In addition, pembrolizumab monotherapy showed prolongation in OS in cases with CPS≥20% and CPS≥1% and proved non-inferiority in the entire population. This result suggests the usefulness of CPS as a biomarker for predicting prognosis when considering administration of pembrolizumab. Based on this, pembrolizumab has been added as an initial treatment option for R/M-HNSCC, which expanded the treatment options. However, the CPS introduced by the founder of KEYNOTE-048 is a subjective evaluation of trained pathologists, and it cannot be said that there is a universal index yet. In the future, it is hoped that evaluations will be standardized using computer-based image analysis tools or like this.

  1. A further point: several useful and important facts are included in the review article, which could be discussed more in detail, as for example: the lower the neutrophil-to-lymphocyte ratio (NLR) and CRP before chemotherapy, the higher the response rate. The archive this aim the influence of immune checkpoint is definitely important. Do the authors see any other option here, especially from the field of treatments for the chronic inflammations?

Thank you for a valuable comment. We have revised the chapter "4.3. Future prospects of ICIs in HNSCC" as the following:

4.3. Future prospects of ICIs in HNSCC

In R/M-HNSCC, we have entered an era in which ICIs are used from the first-line to the second-line treatment (platinum resistance). In recent years, if cytotoxic anti-cancer drug treatment, such as platinum or 5-FU is performed again after using an ICI as a second-line treatment, the immune environment in the tumor tissue changes and the drug sensitivity becomes higher. Its usefulness has also been reported in salvage chemotherapy, which improves sensitivity to tumors [37-39]. And as a biomarker for the response of salvage chemotherapy, it has been reported that the lower the neutrophil-to-lymphocyte ratio (NLR) and CRP before chemotherapy, the higher the response rate [40]. So far, it has been reported that systemic inflammatory responses such as fever (tumor fever) and weight loss may occur in many carcinomas, especially in advanced cases. It has also been shown to have a poor prognosis for patients with a strong inflammatory response [41-43]. NLR and CRP indicate the strength of the inflammatory response, and these markers may also be useful in predicting prognosis in head and neck cancer. In particular, the administration of ICIs changes the micro-immune environment of the tumor, so examining the inflammatory response after administration of ICIs may be a predictor of the effectiveness of salvage chemotherapy. Although more cases still need to be accumulated, NLR and CRP may be useful as prognostic biomarkers. In addition, as a characteristic of immunotherapy, the response rate on advanced cancer is relatively low, but in cases to keep stable disease (SD), the tumor suppression effect is long-lasting (durable response) due to the mechanism mediated by the host immune system, and the prolonging survival is achieved. Superior potential is also shown in the 2-year survival data from the CheckMate-141 trial [44]. In other words, there is a possibility that treatment can be continued for a long period of time while maintaining a certain level of QOL. Furthermore, in recent years, liquid biopsy tests have attracted attention for detecting microsatellite instability and tumor mutational burden. The liquid biopsy tests may be used as a predictor of the response of ICIs [45].

Thus, ICI therapy is a very attractive treatment option and will continue to provide significant benefits to patients with HNSCC. In the future, after predicting the response by liquid biopsy etc., we select an appropriate ICI, and use it in combination with a molecular target drug such as VEGF (vascular endothelial growth factor) inhibitor, or anti-PD-L1 which has already been approved for other cancers such as melanoma. Clinical applications including combined use of anti-PD-1 antibody drug and anti-cytotoxic T-lymphocyte antigen-4 (anti-CTLA-4) antibody drug is expected.

  1. The chapter multi-oncogene panel tests is also interesting, containing up to 324 genes utilized, and finally authors end up still with CTLA-4 and PD-L1 antibodies which are currently used, and are nothing new. If it is like this, also critical remarks could be expected here.

Thank you for a valuable comment. Regarding this point, we made a more critical discussion by adding the topic "Effect of single cell analysis on multi-oncogene panel tests" pointed out by Reviewer 1.

  1. An absolute highlight in the submitted review article is the report about Cetuximab saratolacan, which has a tumor specific potential and is absolutely novel approach. Please provide more details on this especially if it is only usable for oral cancer or also oropharynx, etc. could it be also used during pan-endoscopy or only in ambulant setting?

Photoimmunotherapy with cetuximab salatracan is a useful treatment option that can be widely applied to various HNSCCs, not limited to oral and oropharyngeal cancer, which are EGFR-positive tumors in anatomical areas that are easily exposed to near-infrared light. It is also possible to irradiate with an endoscope, and I think it can be applied to both outpatients and inpatients.

Round 2

Reviewer 1 Report

The authors addressed all the issues that were previously concerned.

Reviewer 2 Report

Dear Authors,

My comments were not taken into account in full.

  1. IJMS is a journal with a significant IF index. Articles should be prepared at a high level. These should be advanced systematic reviews, not a review article. The authors did not change the layout of the paper.
  2. The authors do not write down the names of all the genes in italics.